# Green, Brown, and Gray: Associations between Different Measurements of Land Patterns and Depression among Nursing Students in El Paso, Texas

**DOI:** 10.3390/ijerph17218146

**Published:** 2020-11-04

**Authors:** José Ignacio Nazif-Munoz, José Guillermo Cedeno Laurent, Matthew Browning, John Spengler, Héctor A. Olvera Álvarez

**Affiliations:** 1Faculty of Health and Medicine, Université de Sherbrooke, Sherbrooke, QC J1K 2R1, Canada; 2Department of Environmental Health, Harvard University, Cambridge, MA 02115, USA; memocedeno@mail.harvard.edu (J.G.C.L.); spengler@hsph.harvard.edu (J.S.); 3College of Behavioral, Social and Health Sciences, Clemson University, Clemson, SC 29634, USA; mhb2@clemson.edu; 4School of Nursing, Oregon Health & Science University, Portland, OR 97239, USA; olveraal@ohsu.edu

**Keywords:** greenness, brownness, depression, structural equation models

## Abstract

Background: While greenness has been associated with lower depression, the generalizability of this association in arid landscapes remains undetermined. We assessed the association between depression and residential greenness, but also brownness and grayness among nursing students living in El Paso, Texas (the Chihuahuan desert). Methods: Depression was measured with the Patient Health Questionnaire-9 scale and greenness with the normalized difference vegetation index across three buffer sizes (i.e., 250, 500, and 1000 m). Using data from the National Land Cover Database, two additional measures of land patterns were analyzed: grayness and brownness. Structural equation models were used to assess the relationships of these land patterns to depression and quantify the indirect effects of peer alienation. Results: After adjusting for individual characteristics, at buffers 250 m, greenness was not associated with a decrease in the Incidence Rate Ratios (IRR) of depression (IRR, 0.51; 95% CI, 0.12–2.10); however, grayness and brownness were respectively associated with increases by 64% (IRR, 1.64; 95% CI, 1.07–2.52) and decreases by 35% (IRR, 0.65; 95% CI, 0.42–0.99). At buffer 250 m, peer alienation explained 17.43% (95% CI, −1.79–36.66) of the association between depression and brownness, suggesting a pathway to depression. Conclusions: We did not observe an association between depression and residential greenness in El Paso, Texas. However, we did observe a protective association between brownness and depression and an adverse association with grayness. These results have theoretical implications as they were based on commonly used frameworks in this literature, and adverse association of brownness (and the lack of greenness) and depression was expected.

## 1. Introduction

The connection between people and natural settings matters. Evidence of the protective effects of green natural settings (i.e., greenness) on human health can be traced back to the late nineteenth century [1]. The past twenty years have seen a surge of research that reinforces the protective effects of greenness [2,3,4,5] across a broad range of health outcomes, including cardiovascular health [6], obesity [7], and well-being in children [8,9,10,11], adolescents [12], adults [13], and elderly populations [14].

Multiple studies across the world have also documented a link between greenness and mental health, particularly with depression. One seminal study with residents of London reported that the prevalence of depression was mildly associated with lack of greenness in the built environment [15]. Specifically, individuals living in neighborhoods with few private gardens were 29% more likely to report depressive symptoms than residents living in neighborhoods with numerous gardens. Similar findings were reported in Santiago, Chile; residents were 6% more likely to report depressive symptoms if they lived in neighborhoods with low levels of green elements [16]. In the Netherlands, a higher level of green land cover around the home was associated with a 4% decrease in depression rates [17]. Numerous other studies have documented protective effects on mental health, particularly on depression, of greenness measured with the normalized difference vegetation index (NDVI) [18,19,20,21,22,23,24,25]. NDVI is calculated from remotely sensing imagery from red and infrared wavelengths to measure “greenness,” which indicates healthy green vegetative cover.

Despite this mounting evidence, our understanding of the relationship between natural settings and mental health remains limited for two principal reasons. First, this relationship has been largely studied in territories in which greenness is relatively prominent, leaving the green–depression association unexplored in arid regions with desert landscapes. Studying this association in the desert could reveal the lower boundaries under which greenness may reduce depression. In other words, by considering areas in which green is less prominent, we may be able to identify baseline green values which indicate when a protective factor begins. In addition, the generalizability of how greenness affects mental health must be tested across all climates/vegetative conditions including arid environments, since desertification is occurring globally due to climate change and human activities [26]. Second, the extent literature has tested the potential pathways or mechanisms between natural settings and health is relatively little [27]. There are individual mechanisms and social pathways. Regarding individual mechanisms, restoration theory suggests that individuals exposed to nature—including greenness—facilitates attention restoration [28,29] by supplying the brain an opportunity to regain rest from mental fatigue [30]. Of the possible social pathways identified [31], the capacity of natural settings to encourage social support, aside from air pollution filtration, has been documented by a larger number of studies than any other possible mechanism [27]. Specific to mental health, it has been theoretically suggested that greenness can decrease depression by increasing social support or decreasing peer alienation [32]. Public spaces’ characteristics, through the presence of specific natural surrounding characteristics such as greenness, provide more opportunities to engage in social interaction since encounters to meet relatives, friends, neighbors, co-workers, and strangers are facilitated [33]. Further, the familiarity and associations with these places and social encounters become meaningful and memorable [34].

In the current study, we further develop our understanding of the effects of greenness on depression by evaluating this association using data from the Nurse Engagement and Wellness Study (NEWS) cohort of students in El Paso, Texas, USA [35]. Because these students live in the *Chihuahuan* desert, we explore the generalizability of the claim that greenness could improve mental health outcomes in environments with low baseline levels of greenness. For this, we include continuous measures of “grayness” (i.e., buildings, roads, parking lots, and other impervious surfaces) and “brownness” (arid pervious natural settings without vegetation) to contrast results with greenness. We also respond to the recent call for complex mediation models by Dzhambov and colleagues [27] and employ structural equation models to explore different social pathways, including social support and peer alienation, between greenness and depression. With this unique geographic context and analytical frame, we postulate how greenness and/or other residential land patterns (grayness or brownness) may impact depression by increasing social support or reducing peer alienation.

## 2. Materials and Methods 

### 2.1. Study Population

The analysis was carried out as part of the NEWS, described in detail previously [35]. Briefly, the NEWS is a prospective cohort study of nursing students directed by a team of nursing, social, and environmental health researchers. Participants were female or male students between 18 and 60 years old enrolled in a Bachelor of Science in Nursing program. Participants were recruited via emails, posters, flyers, media outlets and in-class information sessions and they consented to be part of the study. Self-administrated questionnaires were used to obtain information on socio-economic characteristics, ethnicity, lifestyle factors, environmental exposures, and physical and mental health. The questionnaires were conducted between May 2015 and December 2018. Students responded the questionnaires within the first semester in the Bachelor of Science in Nursing program. Participants who dropped out of the Bachelor of Science in Nursing program, transferred to another university, or failed to graduate from the program were removed from the sample. The study was approved by institutional review boards at the University of Texas at El Paso (857149–1) and Harvard University (16–0080).

### 2.2. Outcome

Depression was measured with the patient health questionnaire-9 (PHQ-9), a multipurpose instrument for screening, diagnosing, monitoring, and measuring the severity of depression consisting of nine items [36]. The range of values of this instrument is 1 to 27. The PHQ-9 has been consistently validated for the general population in different countries [37,38]. Since analyses of depression can be affected by how this variable is operationalized [39], we also applied an ordinal measurement of depression. We used the following three categories: minimal depression (0–4), mild to moderate depression (5 to 14), and moderate severe to severe depression (15 to 27) [40]. 

### 2.3. Exposures

Averages of three land patterns were calculated in 250 m, 500 m, and 1000 m Euclidean buffers centered on each participant’s geocoded address of residence. This range of buffer size (250 m to 1000 m) are associated with the pathways being tested [41], whereas larger buffer sizes show a distance decay function with depressive symptoms [24] and are less robust than buffers between 300 and 500 m [42,43]. Three land patterns were tested individually: greenness, grayness, and brownness.

Greenness was the mean of the NDVI values from 30 m2 pixels calculated from cloud-free Landsat 7 satellite imagery during a single summer day in 2016 [44,45]. The possible range of NDVI values is −1 to 1 with higher values indicating the presence of vegetation and negative numbers indicating bodies of water. Because negative values were present in less than 5% of the study region, these were reclassified as missing data to avoid blue space (i.e., the Rio Grande river and irrigation canals) degrading greenness values and confounding results. 

Grayness was the percentage of concrete, buildings, and natural rock (i.e., contiguous surface outcroppings, such as “slickrock”) that do not allow water infiltration. Higher values indicate higher proportions of impervious than pervious surfaces. Data were obtained from the National Land Cover Database, which was developed by the Multi-Resolution Land Characteristics Consortium (www.mrlc.gov). The National Land Cover Database displays 80% land cover classification accuracy [46].

Brownness was calculated from the residual land cover unassigned to greenness or grayness as defined by the following equation.
Brownness = 1.00 − Greenness − Grayness(1)
The resulting measure of brownness consisted of pervious non-biologic land covers not captured by impervious land covers (i.e., uniform or mixed sand, silt, clay, soil, mud, and rocks broken into pieces) as well as not/low-green biologic material not captured with NDVI (i.e., unhealthy/dead shrubs and trees, lichen, and biological soil crusts) [47]. Visual inspection of the three land cover classes used here (brown, green, and gray) showed that high gray areas were highly developed with buildings, roads, and parking lots; high brown areas were largely what a reasonable person would expect to be desert; and higher green areas (max of 0.11 in our sample) had more street/yard/riparian trees and shrubs than other areas of the study region (images of different measurement of land patterns are available in Appendix A). Although grayness could have included natural rock outcroppings based on the data processing from the NLCD, visual inspection identified high values only in highly developed areas—thus, grayness was indeed a measure of the built environment for residents living in El Paso, TX.

### 2.4. Mediators

Two variables were considered for exploring potential pathways. 

Social support was an index derived from seven questions [48] and ranged from 7 to 32. Higher values indicated stronger social support for the individual. (File S1 gives the list of questions and answers.) 

Peer alienation was derived from a sub-scale of the Peer Attachments index, which was derived from the 25 peer-related questions from the Inventory of Parent and Peer Attachment [49] and ranged from 7 to 35. Lower values indicated weaker attachment to peers. (See Appendix A gives the list of questions and alternatives.)

### 2.5. Covariates

Our critical review of the literature identified other five relevant covariates [50,51,52,53,54,55,56,57,58,59]. Adverse Childhood Experiences was measured using the ACE questionnaire [54]. Values of this variable go from 0 to 10. A healthy-food index was created from six items, which asked participants how often they typically consumed fruits, vegetables, white meat, fish and shellfish, whole grain food, or nuts. The scale reliability coefficient (Cronbach’s alpha) was 0.74, and values of this index ranged from 1.35 to 6.5. Higher values indicated healthier food habits. A physical-index activity was created from nine items asking questions regarding physical activity such as walking, swimming, and jogging. The scale reliability coefficient (Cronbach’s alpha) was 0.77, and values ranged from 0 to 7. Higher values indicated more physical activities (See Appendix A gives the list of questions and alternatives). Body mass index was obtained from each participant’s height and weight reported at the day of the interview. Household income was self-reported and based on ten categories: less than $5000; $5000 through $11,999; $12,000 through $15,999; $16,000 through $24,999; $25,000 through $34,999; $35,000 through $49,999; $50,000 through $74,999; $75,000 through $99,999; and $100,000 or greater. Age and sex were also measured.

### 2.6. Analyses

We applied generalized structural equation models [60] to first assess the magnitude of the direct associations between depression and land cover patterns, and then measure the proportion of social support or peer alienation explaining the total association between depression and our exposures of interest. Furthermore, we adjusted all the models with a separate set of sufficient confounders for each co-variate using a priori knowledge of the relationship between greenness and depression [32]. This strategy was adopted to avoid the “Table 2 fallacy” [61], which occurs when investigating multiple predictors using a single multivariable model. Such models do not provide unbiased total effect estimates for the predictors of interest, since some of the covariates may be on the causal pathway between the predictor of interest and outcome. Our hypothesized structural model linking greenness with social support and peer alienation, and ultimately depression, is given in Appendix A, Appendix A. 

To assess direct associations, several modeling approaches were run to account for the possibility of non-linear associations or non-normally distributed outcomes. Specifically, we used Poisson, negative binomial, and gamma regression models to assess the continuum measurement of depression [62]. The outcome was log transformed in these models to compare results with ordinary least squares regression results [63]. To model the ordinal measurement of depression, we used Poisson, negative binomial, ordinal probit, Bernoulli probit, ordinal logistic, and Bernoulli logistic models. These functions allowed us to determine which models best fitted the data and correctly estimate the direct associations between depression under its two operationalizations (continuous or ordinal) and land cover patterns. We report results using Incidence Rate Ratios (IRR). These are rates of two different groups, in which the rate of a group of interest is divided by a comparison group. An IRR equal to 1.0 indicates equal rates among both groups, whereas an IRR greater than 1.0 indicates a higher risk for the comparison group and an IRR lower than 1.0 a decreased risk for the group of interest [64].

In Appendix A, Appendix A, we provide results for multicollinearity tests and bivariate correlation plots among the variables of interest. To measure the proportion of the association mediated by social support or peer alienation, we divided their indirect association by the total effect. In Appendix A, Appendix A, we provide both standardized root mean squared residual and coefficient of determination results regarding the structural models. All statistical analyses were performed with Stata 16 [65] and all geospatial data processes and buffer creations were performed in ArcMap 10.4.1 (Esri, Redlands, CA, USA) [66].

## 3. Results

Depression (PHQ-9) scores did not follow a normal distribution (Table 1), and thus distributions such as Poisson, negative binomial or gamma were better suited to examine its variation under the continuous operationalization. Characteristic of an arid region, greenness was very low with mean NDVI values of 0.11. Grayness mean values were approximately three times higher than greenness values, approximately 0.35. As expected for the arid study region, we observed the highest levels of the three land patterns for brownness, which displayed means over 0.5. The sample was composed primarily by women (78.95%) and the average age was 25 years.

### 3.1. Greenness and Depression

Results from the 36 generalized structural equation models (4 functions × 3 buffers × 3 measures of residential land) were grouped by measure of residential land cover (e.g., greenness, grayness, or brownness) and are displayed in Figure 1, Figure 2 and Figure 3. Results are reported as IRR adjusted for all covariates, and in the X axis of each figure, ranges of IRR values are represented. The direct association of each covariate in each model is presented in Appendix A, Appendix A. 

Figure 1 shows that higher values of greenness are associated with lower values of depression. In terms of best fit models for depression as a continuous variable, it was observed that the normal distribution better fits the data with depression scores logged transformed. In each model with the Normal distribution, Akaike information criterion (AIC) and Bayesian information criterion (BIC) values were consistently the lowest ones. The IRRs for depression and greenness for this distribution is 0.51 (95% confidence interval (CI): 0.12, 2.10) at 250 m, 0.54 (95% CI: 0.13, 2.29) at 500 m, and 0.58 (95% CI: 0.13, 2.59) at 1000 m. For depression as ordinal variable, the Bernoulli–Logit distribution best fit the data since the AIC and BIC values for this distribution were the lowest. The IRRs for this distribution are 0.20 (95% CI: 0.00, 14.33) at 250 m, 0.16 (95% CI:0.09, 15.94) at 500 m, and 0.09 (95% CI: 0.06, 17.84) at 1000 m.

### 3.2. Grayness and Depression

Results for grayness models are given in Figure 2. For the depression as continuous variable, the IRR effects are 1.64 (95% CI: 1.07, 2.52) at 250 m, 1.62 (95% CI: 1.05, 2.51) at 500 m, and 1.52 (95% CI: 0.99, 2.34) at 1000 m for the normal distribution models. For the ordinal outcome, the IRR results are 2.58 (95% CI: 0.77, 8.61) at 250 m, 1.68 (95% CI: 0.77, 9.09) at 500 m, and 2.44 (95% CI: 0.72, 8.22) at 1000 m for the Bernoulli–Logit distribution models.

### 3.3. Brownness and Depression

Results for brownness models are given in Figure 3. For the depression as continuous variable, the IRR pooled effects are 0.65 (95% CI: 0.42, 0.99) at 250 m, 0.65 (95% CI: 0.41, 1.01) at 500 m, and 0.68 (95% CI: 0.44, 1.06) at 1000 m for the normal distribution models. For the ordinal outcome, the IRR results are 0.45 (95% CI: 0.14, 1.53) at 250 m, 0.44 (95% CI: 0.13, 1.53) at 500 m, and 0.49 (95% CI: 0.14, 1.70) at 1000 m. for the Bernoulli–Logit distribution models.

### 3.4. Social Pathways Between Depression and Environmental Exposures

Next, we report the results of the social support and peer alienation pathways between greenness, grayness, and brownness and depression. More specifically, we report the proportion explained by these two mediators when assessing the association between depression and environmental exposures. The pathways of interest are represented by red arrows in Appendix A in Appendix A. Results of Appendix A were used to assess the proportion explained by social support and peer alienation after measuring the total effect between depression and the environmental exposures introduced at 250 m buffer.

Figure 4 shows the results corresponding to the two analyzed social pathways between greenness, grayness, and brownness and depression as continuous variable per model (results for depression as categorical variable available in Appendix A
Appendix A). Social support (Figure 4a) and peer alienation (Figure 4b) did not mediate the total effect between depression and greenness. For social support, the effect of the proportions at 250 m is −2.26% [−4871, 44.20]. For peer alienation, we observe that the effect of the proportions at buffer 250 m to be 6.54% [−32.89, 45.97] for depression as continuous variable. Like the results for greenness, social support (Figure 4c) did not show a mediation effect; its effect at 250 m is 4.57% [−12.65, 21.79]. Peer alienation (Figure 4d) seems to mediate the grayness–depression association at 250 m by 17.43% [−1.79, 36.66]. Like the results for greenness and grayness, social support (Figure 4e) did not mediate these associations; its effect at 250 m is 3.42% [−16.18, 23.02]. Whereas, peer alienation (Figure 4f) seems to mediate the association between brownness and depression. Increases in brownness were associated with decreases in peer alienation, which in turn were associated with lower incidences of depression. Its mean effect at 250 m is 18.54% [10.40, 33.02].

## 4. Discussion

We observed negative but null associations between depression and residential greenness measured with NDVI in a cross-sectional sample of nurse students in El Paso, Texas. We also found a protective association between brownness and depression as well as an adverse association of grayness. These results suggest that in desert regions, the predominant physical environmental elements may be associated with depression. We also noticed that peer alienation partially mediated the link between depression and brownness and grayness.

Our results do not support a direct association between depression and greenness in the desert environment of El Paso among students, who are at risk of depression [35]. We believe that the lack of an association between greenness and depression in our study could be a result of relatively low exposure levels to greenness in El Paso, where the median NDVI within buffers of 250 m is 0.11 SD 0.03 (Inter Quartile Range 0.09; 0.11). This level is considerably below USA’s mean NDVI reported by James and colleagues [3] at the same buffer size of 0.47 SD 0.12. The lack of association between greenness and depression observed here may suggest the existence of a given threshold for greenness to have a positive association with depression, which is unmet in El Paso. Conducting future research to confirm the lower threshold level for which greenness may affect depression is an important task for two reasons: first, depression has been shown to be a key mediator between greenness and overall mortality in the US [3], and second, as desertification and human exposure to desert landscapes increases globally, understanding the minimal levels of greenness that we need to preserve is essential.

The observed protective association of brownness with depression in the continuous outcome could be explained by the attention restoration theory discussed in the ethnobiology literature [28,67]. Attention restoration theory suggests that perceptual fluency displayed in overall nature—green or otherwise—facilitates attention restoration [29] providing the brain an opportunity to recover from mental fatigue [30]. In this case, the fractal patterns associated with the prominent brown elements of the dessert landscape may reduce stress by facilitating the perception of a steadier repetition of visual information. It is also plausible that participants of our study, who are mostly long-term residents of the region, have become accustomed to the brown natural elements in El Paso and hence find it easier to assess safety cues in that type of environment and hence counteract the fight or flight response triggered by the acute social stressor [68]. In our study, the adverse association of grayness with depression also supports this interpretation since we observed considerable increases of depression for those individuals more exposed to impervious surfaces such as concrete or buildings. While we observed that grayness measured the built environment, it is possible that in some cases grayness also included natural impervious rock outcroppings. Such situations would be uncommon and are unlikely to contribute meaningfully to the direction or effect size of the associations observed here.

In terms of pathways between residential land patterns and depression, two findings associated with social factors were studied, namely the null associations for social support and the mediation of peer alienation. While this study did not find evidence that social support significantly mediates the effect between land pattern and depression, we observed an association of similar magnitude to that described in a longitudinal study of 4118 older adults across the United States with regard to stress and greenness [20]. In their analyses, social support mediated only 4.38% of the association between stress and greenness. In our current study, social support mediated 2.26% of the association between depression and greenness, yet our results were not significant, which may indicate our analyses were underpowered. Conversely, our results suggest that peer alienation explains about 18% of the association between environmental exposures and depression. More precisely, depression increases if peer alienation increases, which in turn happens when brownness decreases or grayness increases. We are unaware of other studies of greenness and depression, or other human health outcomes, that have evaluated this specific concept. However, our findings match those of a study of 10,089 Dutch residents that found decreases in social isolation partially mediated the relationship between greenness and self-reported risk of psychopathology [69]. While more research is needed to validate the mediating role of alienation or isolation between natural settings and mental health, the parallel findings from these two studies with vastly different contexts (temperate Europe and American desert) and the lack of null findings from other studies to the best of our knowledge hint that this effect may be generalizable to other contexts.

While this study presents novel findings regarding how brownness should be more explicitly analyzed and considers possible mechanisms at play, the analysis has a few limitations. As this is a cross-sectional study, the ability to draw causal inferences was limited since reverse causation cannot be addressed. It is important to acknowledge that the low number of participants in the study should temper our interpretation. However, all the control variables used in the analysis were similar in magnitude and direction as the literature in depression has suggested. In addition, these results may not be generalizable to other populations living in desertic zones because our sample was composed of university students. While the NDVI is a reliable measure of green space [25], this scale does not distinguish between different types of green areas such as tree canopy or parks, which may be related to the different magnitudes of the tested direct and indirect associations, nor does it address quality of green spaces. In this regard, it is also important to acknowledge that our research needs to be improved with both theoretical and methodological tools used in spatial cognition studies to properly address how a brown environment affects cognition as well as mental health. In terms of exposure time or seasonality to greenness, grayness or brownness, it was assumed that members of our sample were similar in this regard, yet some students may have been exposed longer to specific surroundings, with more or less green, gray or brown, or over time the amount of greenness may have changed in their neighborhoods, therefore affecting measurements of exposure to land patterns. Specifically, depression was assessed between May 2015 and December 2018, whereas NDVI was measured in July 2016. This measurement of greenness was considered representative of the long-term exposure of participants. This was considered reasonable as the diversity of vegetation in the desert region of El Paso is small and hence the spatial variation of greenness is limited as well. In other words, if a participant had a high NDVI value in July 2016, that person is expected to be exposed to higher levels of greenness—at home—throughout the year, than someone that lived in a part of El Paso with lower NDVI values in July 2016. From this perspective, we were examining the association between long-term of exposure to residential greenness and depression severity among young residents at a given point in time (e.g., first semester of their nursing program). These assumptions have several limitations. For instance, there is a possibility that spatial variability of greenness might have seasonal changes that affect the interpersonal assessment of exposure to greenness. Similarly, exposure to greenness away from home, such as at school or work, would affect the exposure assessment [70]. Our results should be considered as preliminary, and more research needs to be conducted with comprehensive longitudinal assessment of exposure to confirm our findings.

## 5. Conclusions

The results of this study suggest that greater presence of brown space in neighborhoods embedded in desert-like environments may be associated with lower levels of depression, even when controlling for individual-level confounders. Future studies could employ longitudinal data to further address neighborhood self-selection, reverse causality, and how the desert may impact perceptual fluency and its association with depression. Furthermore, more avenues of research should be explored to better understand both social pathways and individual mechanisms between brownness, depression, and other mental health outcomes [71]. In this regard, relationships between social contact and physical activity in arid zones are particularly important. For instance, future studies could investigate to what extent arid zones integrate ad-hoc public infrastructure to improve social encounters as well as activities such as walking. Careful attention to temperature should be acknowledged. In terms of mechanisms underlying associations between land covers and depression, more analyses are required to directly test propositions associated with stress reduction theory [72] and attention restoration theory [28]. Efforts in these directions will strengthen our ability to make causal inferences that could facilitate specific efforts to modify or reorganize desert environments in which individual live in order to benefit their mental health.

## Figures and Tables

**Figure 1 ijerph-17-08146-f001:**
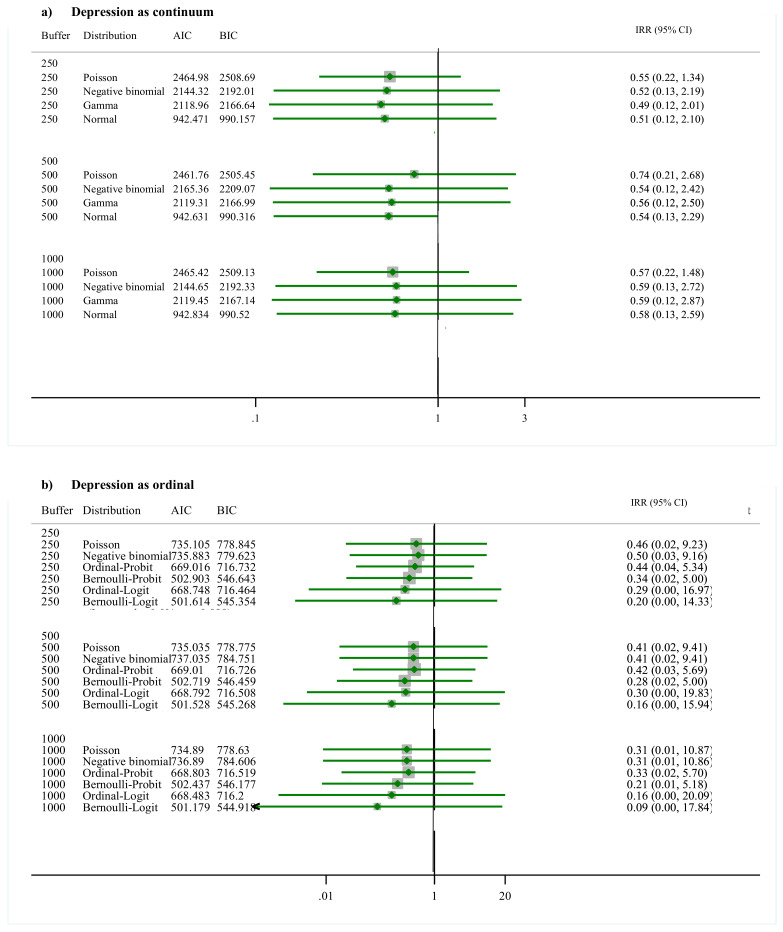
Direct association between depression (**a**) continuum and (**b**) ordinal variable and greenness summarized at buffer sizes of 250 m, 500 m, and 1000 m. *Note*: Buffer indicates distances in meters around the geocoded address of residence of each study participant. Distribution indicates the distribution in which depression was modeled. When the normal distribution was assumed, the values of depression were log transformed. AIC corresponds to the Akaike information criterion results and BIC to the Bayesian information criterion results. IRR are incidence rate ratios. CI is confidence interval. All models were adjusted by Age, Sex, Healthy-food index, Physical-Activity index, Body mass index, Household income, Adverse Childhood Experiences, Social support index, and Peer alienation index.

**Figure 2 ijerph-17-08146-f002:**
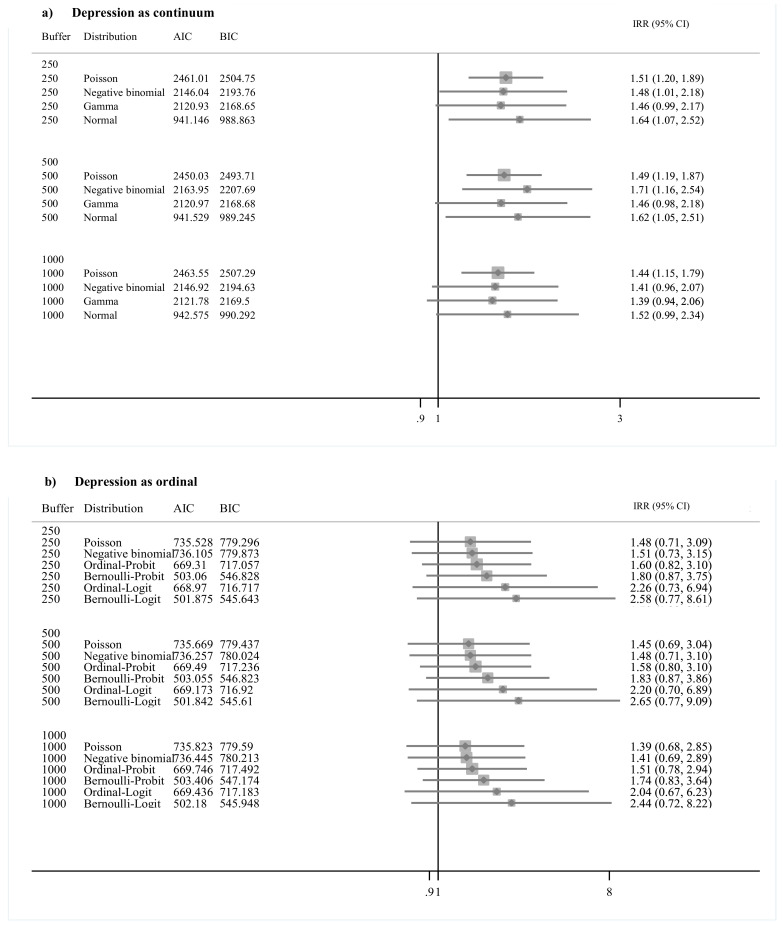
Direct association between depression (**a**) continuum and (**b**) ordinal variable and grayness summarized at buffer sizes of 250 m, 500 m, and 1000 m. *Note:* Buffer indicates distances in meters around the geocoded address of residence of each study participant. Distribution indicates the distribution in which depression was modeled. When the normal distribution was assumed, the values of depression were log transformed.

**Figure 3 ijerph-17-08146-f003:**
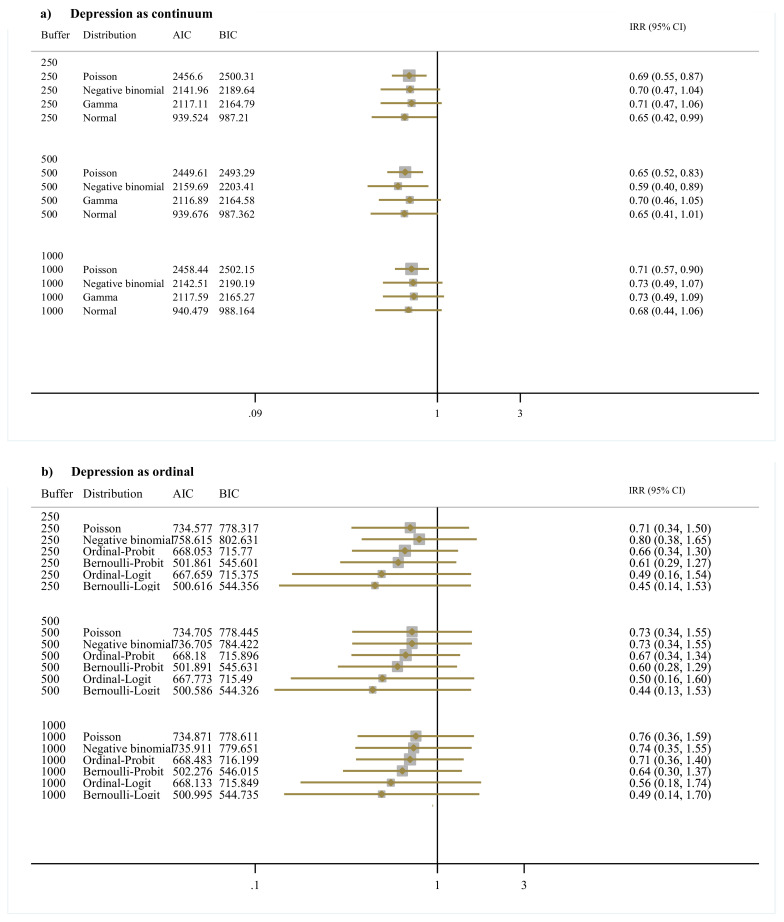
Direct association between depression as (**a**) continuum and (**b**) ordinal variable and brownness summarized at buffer sizes of 250 m, 500 m, and 1000 m. *Note:* Buffer indicates distances in meters around the geocoded address of residence of each study participant. Distribution indicates the distribution in which depression was modeled. When the normal distribution was assumed the values of depression were log transformed. AIC corresponds to the Akaike information criterion results and BIC to the Bayesian information criterion results. IRR are incidence rate ratios. CI is confidence interval. All models were adjusted by Age, Sex, Healthy-food index, Physical-Activity index, Body mass index, Household income, Adverse Childhood Experiences, Social support index, and Peer alienation index.

**Figure 4 ijerph-17-08146-f004:**
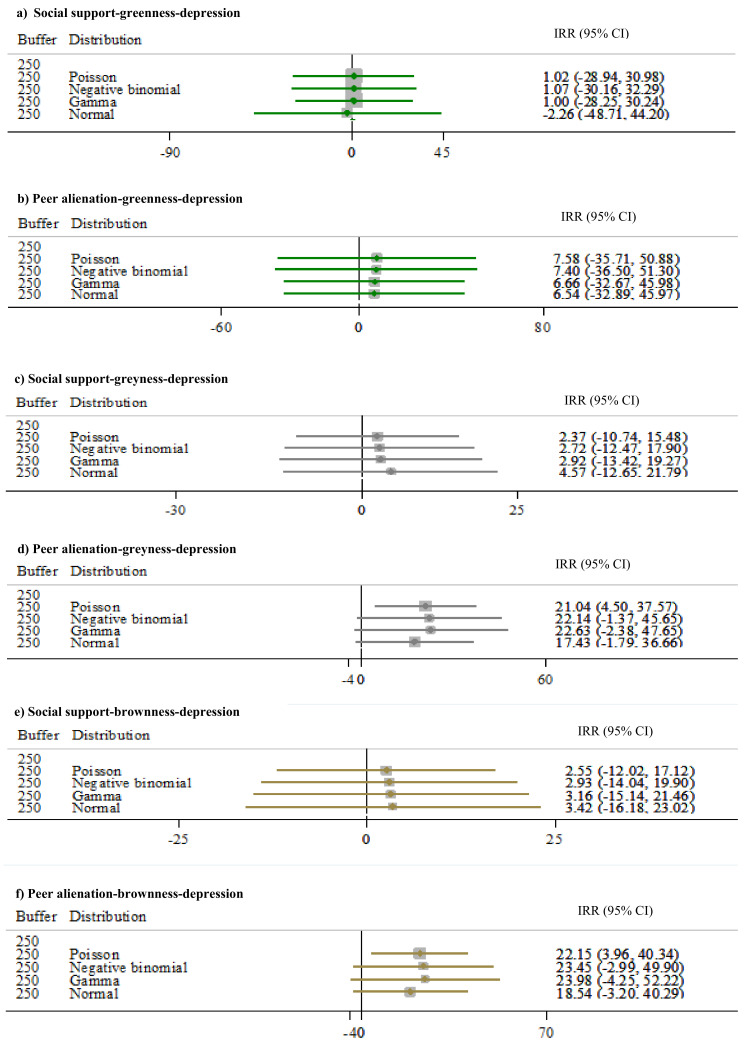
Social support and peer alienation as pathways between greenness, grayness, and brownness and depression (as continuum), in 250 m buffer. (**a**) Social support–greenness–depression. (**b**) Peer alienation–greenness–depression. (**c**) Social support–grayness–depression. (**d**) Peer alienation–grayness–depression. (**e**) Social support–brownness–depression. (**f**) Peer alienation–brownness–depression. IRR (95% CI). *Note:* Buffer indicates distances in meters around the geocoded address of residence of each study participant. Distribution indicates the distribution in which depression was modeled. When the normal distribution was assumed, the values of depression were log transformed. IRR are incidence rate ratios. CI is confidence interval. All models were adjusted by Age, Sex, Healthy-food index, Physical-Activity index, Body mass index, Household income, Adverse Childhood Experiences, Social support index, and Peer alienation index.

**Table 1 ijerph-17-08146-t001:** Description of Sample (*n* = 393).

Variables	%	Mean	Std. Dev.	Min	Max
Outcome					
Depression (counts) ^a^		6.41	5.23	1	27
Depression (log)		1.51	0.88	0	3
Depression categorical					
Minimal depression	45.50				
Mild to moderate depression	40.65				
Moderate severe to severe depression	13.86				
Exposures					
NDVI ^b^					
at 250 m		0.11	0.03	0.04	0.28
at 500 m		0.11	0.03	0.05	0.25
at 1000 m		0.11	0.03	0.06	0.26
Grayness					
at 250 m		0.40	0.17	0.00	0.82
at 500 m		0.37	0.17	0.00	0.74
at 1000 m		0.36	0.17	0.00	0.90
Brownness					
at 250 m		0.50	0.17	0.14	0.94
at 500 m		0.52	0.17	0.18	0.92
at 1000 m		0.53	0.17	0.03	0.91
Covariates					
Adverse Childhood Experiences		2.02	2.01	0	10
Social support index		27.53	4.89	7	32
Peer alienation index		15.12	4.36	7	35
Healthy-food index		3.89	0.76	1	6
Physical-index activity		1.77	1.05	0	7
Body mass index		25.72	5.37	14.10	47.30
Income		6.22	2.34	1	10
Age		25.15	6.26	18	55
Sex (% women)	78.95				

^a^ Skewness and Kurtosis values for depression were 1.43 and 5.20, respectively. The joint test for these measures was X^2^ 82.77 (*p* < 0.000); ^b^ NDVI indicates the normalized difference vegetation index.

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
