# Peer review of "Green, Brown, and Gray: Associations between Different Measurements of Land Patterns and Depression among Nursing Students in El Paso, Texas"

_ijerph, 2020, doi:10.3390/ijerph17218146_

Round 1
Reviewer 1 Report
In terms of the relationship between the surrounding greening environment and mental health, this paper can draw attention to the degree of green space by considering not only green space but also grey and brown.
However, in some parts of this paper, it does not seem to be kind to the readers who can be students or other related field researchers.
To improve this paper more solidity, I'd like to suggest some minor comments.
- There are too many abbreviations in this paper.
- (line 56 to )I understand the authors' argument regarding the relationship between natural settings and mental health. I'm sorry, but what does mean the 'mechanistic pathways' of second reasons in this context?
- (appendix B) Unfortunately, I cannot understand this SEM without any coefficient, and what is the meaning of the line color? AND, why isn't there a 500m result on the direct effect of predicting depression for four distributions? (appendix table.B)
- This paper needs to describe in more detail the research design that sets up the geocoded address of each patient and collect the corresponding answers. There is no spatial background and set-up criteria considered in the research design and their current status.
- (line 193)what is mean the 'GSMs' that seem to be abbreviations?
- check the status of Figure2 and Figure3.
- (discussion section)In this paper, I'm wondering that the author is conducting such discussion even though no discourse such as 'spatial', 'spatial and cognitive', and etc about brown space are presented.
To be honest, it is doubtful whether the failed contents submitted somewhere were copied and pasted into the mdpi word file without any trimming. Therefore, I felt that the overall reading was not smooth.
Author Response
Reviewer 1
In terms of the relationship between the surrounding greening environment and mental health, this paper can draw attention to the degree of green space by considering not only green space but also grey and brown.
However, in some parts of this paper, it does not seem to be kind to the readers who can be students or other related field researchers.
To improve this paper more solidity, I'd like to suggest some minor comments.
- There are too many abbreviations in this paper.
We added a sub section of abbreviations at the end of the paper and reduce its use in three concepts. Please see page 14.
2. (line 56 to )I understand the authors' argument regarding the relationship between natural settings and mental health. I'm sorry, but what does mean the 'mechanistic pathways' of second reasons in this context?
Agree we deleted the mechanistic because it creates confusion.
3. (appendix B) Unfortunately, I cannot understand this SEM without any coefficient, and what is the meaning of the line color? AND, why isn't there a 500m result on the direct effect of predicting depression for four distributions? (appendix table.B).
We added 500m results and explain the meaning of colors in the figure.
4. This paper needs to describe in more detail the research design that sets up the geocoded address of each patient and collect the corresponding answers. There is no spatial background and set-up criteria considered in the research design and their current status.
The reviewer is correct that there is no clear rationale for the buffer sizes used for summarizing NDVI values around the homes of participants. This is because there is no agreed upon rationale in the literature. In response we decided to use a set of buffer sizes that capture the range of the most common values used in the greenness literature including a value seen already to be associated with depressive symptoms in the literature. We described this rationale in the first paragraph of section 2.3. Lines 116-117
This range of buffer size (250 m to 1000 m) are associated with the pathways being tested (Davand et al. 2014), whereas larger buffer sizes show a distance decay function with depressive symptoms (Browning et al. 2019). Three land patterns were tested individually: greenness, greyness and brownness.
5. (line 193)what is mean the 'GSMs' that seem to be abbreviations?
We spelled the meaning to generalized structural equation models
6. check the status of Figure2 and Figure3.
The status of these two figures were corrected
7. (discussion section)In this paper, I'm wondering that the author is conducting such discussion even though no discourse such as 'spatial', 'spatial and cognitive', and etc about brown space are presented.
Thank you. This comment is extremely important. We have acknowledged this to be a limitation by referring to the discipline of spatial cognition. Please refer to lines 376-378
8. To be honest, it is doubtful whether the failed contents submitted somewhere were copied and pasted into the mdpi word file without any trimming. Therefore, I felt that the overall reading was not smooth.
Thank you for this comment we have re read, the manuscript twice and improved when writing was not clear.
Reviewer 2 Report
It is an interesting research exploring the relationships between different landscape patterns with depression. The following are some suggestions for improving the pertinence and readability. First, the title “Some green is better than nothing” seems not consistent to one of the main results of this research that “We did not observe an association between depression and residential greenness in El Paso, Texas”. How about “Is any green better than nothing?” or other proper statements? Second, the Result section just simply repeats the statistical results presented in corresponding figures, which is not detailed enough. This section should also encompass the authors’ explanations for each finding (i.e. statistics in this research). Furthermore, associations with previous researches for each result can be developed to provide the consistence or differences so that readers can evaluate your results with a broader context. Third, please double check the typos, grammars, etc. There are some minor mistakes, such as line 132-133, “Social support …… ranged from 7 to 32” instead of “7 to 35”; line 137, “Higher values indicated weaker attachment to peers”, I suppose it should be “stronger attachment” due to your questions; and line 141, “Our critical …… seven relevant covariates”, but I just find five covariates in the following text.Author Response
- It is an interesting research exploring the relationships between different landscape patterns with depression. The following are some suggestions for improving the pertinence and readability. First, the title “Some green is better than nothing” seems not consistent to one of the main results of this research that “We did not observe an association between depression and residential greenness in El Paso, Texas”. How about “Is any green better than nothing?” or other proper statements?
Thank you we have changed the title to something more accurate.
- Second, the Result section just simply repeats the statistical results presented in corresponding figures, which is not detailed enough. This section should also encompass the authors’ explanations for each finding (i.e. statistics in this research). Furthermore, associations with previous researches for each result can be developed to provide the consistence or differences so that readers can evaluate your results with a broader context.
We appreciate this comment. We were very succinct in the result section because we are reporting more than 36 equations. This can be very overwhelming to readers. In terms of previous researchers we prefer to discuss results in this way in the discussion.
- Third, please double check the typos, grammars, etc. There are some minor mistakes, such as line 132-133, “Social support …… ranged from 7 to 32” instead of “7 to 35”;
There is a confusion since social support has that range (7-32) whereas peer alienation had another range (7 to 35). We believe this may have created the confusion. We have rerun the analyses of social support and the scale for our sample corresponds to the aforementioned range.
- line 137, “Higher values indicated weaker attachment to peers”, I suppose it should be “stronger attachment” due to your questions;
Thank you for this observation. Indeed it is as the reviewer mentioned. We have replaced the word to lower values.
- and line 141, “Our critical …… seven relevant covariates”, but I just find five covariates in the following text.
This observation is correct. Social support and Peer alienation were part of the review and this is why we mentioned seven. We have changed to five. Thank you.
Reviewer 3 Report
This study addresses an important and as yet overlooked issue: do non-green (or blue) natural spaces confer the same mental health benefits that evidence suggests green spaces do? This study is valuable in that it begins to address that question and will provide inspiration for others to further develop research in this area. This question is also valuable in that it will help to unpick the mechanisms by which benefits are conferred: do they arise because a space is green and/or natural and/or because, for example, the space provides opportunities for people to socialise. My main points are that the messaging seems inconsistent throughout the paper, from the title to the abstract and through to the discussion. The results and figures could be presented more clearly and key terms spelt out. This paper is likely to be of interest to those from environmental as well as health disciplines and as such should be accessible to both. The conclusions section could be stronger and offer a broader range of opportunities for further research.
Specific points are as follows:
The title does not seem to match well with the message in the text. Isn’t the point that “brown” natural spaces could, in arid landscapes, provide the mental health benefits that green spaces provide in landscapes where natural spaces are predominantly green? Also, you say in the abstract (and e.g. line 339) that you don’t find a relationship between greenness and depression which seems to contradict the title.
Line 13: Residential greenness rather than just greenness
Line 21: “at buffers 250 m” is unclear
Line 26: this seems to contradict the point in line 21/22 where you say there was an association between greenness and incidence of depression.
Line 29: “an adverse” not “and adverse”
Line 56. The value of this study comes from trying to begin to understand the mechanisms by which different spaces provide benefits for mental health. As the authors say, much of the previous research has focused on green (and to a lesser extent blue) spaces and so have ignored other types of habitat. By looking a brown, green and grey spaces, it is possible to start to understand whether mental health benefits come from a space being green, being “natural” rather than man made (which could be green or brown) or a public space that allows people to socialise (which could be grey, green or brown). I think some discussion of this and other mechanisms by which mental health benefits could be derived from natural spaces would be helpful here. Attention Restoration Theory is mentioned in the Discussion but could be introduced here. The nature connectedness literature could also be explored.
Line 59. “studying this association….” Please explain this point more clearly. Not sure what is meant by “lower boundaries”
Line 100.When was the survey deployed? How does this match with when NDVI data were derived from? What are the implications of this?
Line 128. Some justification needed as to why this is an appropriate measure of “brownness”. What types of land use/cover does this include? You mention above “arid” natural spaces but could it also include some types of manmade spaces?
Line 192. Greenness missing an ‘s’
Line 194. Does this mean that non-residential areas within the buffers were excluded from the calculations?
Line 195. Some guidance on how to interpret IRR values and what they represent would be useful.
Figures 1-3. Some problems with the formatting of the figures – alignment issues and blank spaces appear over different areas of the figures obscuring the text. What is the x axis? IRR? Again, some guidance on what these values mean would be helpful. Denote as (a) continuous and (b) ordinal variables for clarity. I’m not sure what the diamond shapes represent. Figure 3 legend should say brownness not greyness.
Line 259. Proportion of what? Variation?
Figures 4-6. Again some formatting problems as explanation of IRR would be helpful. Could the 3 separate figures be consolidated into one to avoid repetition, save space and allow for easier comparisons?
Line 336. Greyness not grayness (and in other places in the manuscript). As mentioned above, a description of the land cover types the brownness measure represents would be helpful. This would help link with the arguments related to spending time in natural environments below.
Line 351. Or is preserving greenness less pressing if, as you have shown, brown spaces could provide similar benefits?
Line 365. How could the natural rock included in the greyness category affect this interpretation?
Line 400. I would have liked to have seen a broader discussion of the many avenues this research could go in, specifically unpicking the mechanisms by which green and/or natural environments provide mental health benefits.
Author Response
- This study addresses an important and as yet overlooked issue: do non-green (or blue) natural spaces confer the same mental health benefits that evidence suggests green spaces do? This study is valuable in that it begins to address that question and will provide inspiration for others to further develop research in this area. This question is also valuable in that it will help to unpick the mechanisms by which benefits are conferred: do they arise because a space is green and/or natural and/or because, for example, the space provides opportunities for people to socialise. My main points are that the messaging seems inconsistent throughout the paper, from the title to the abstract and through to the discussion. The results and figures could be presented more clearly and key terms spelt out. This paper is likely to be of interest to those from environmental as well as health disciplines and as such should be accessible to both. The conclusions section could be stronger and offer a broader range of opportunities for further research.
Thank you for this comment and for capturing the overall idea of this research effort.
Specific points are as follows:
- The title does not seem to match well with the message in the text. Isn’t the point that “brown” natural spaces could, in arid landscapes, provide the mental health benefits that green spaces provide in landscapes where natural spaces are predominantly green? Also, you say in the abstract (and e.g. line 339) that you don’t find a relationship between greenness and depression which seems to contradict the title.
We have modified the title.
- Line 13: Residential greenness rather than just greenness
We added residential
- Line 21: “at buffers 250 m” is unclear
We changed this by adding “across three buffer sizes”
- Line 26: this seems to contradict the point in line 21/22 where you say there was an association between greenness and incidence of depression.
We would add the “non-significant” to clarify the quality of this association however we have added “not associated”
- Line 29: “an adverse” not “and adverse”
We changed the word
- Line 56. The value of this study comes from trying to begin to understand the mechanisms by which different spaces provide benefits for mental health. As the authors say, much of the previous research has focused on green (and to a lesser extent blue) spaces and so have ignored other types of habitat. By looking a brown, green and grey spaces, it is possible to start to understand whether mental health benefits come from a space being green, being “natural” rather than man made (which could be green or brown) or a public space that allows people to socialise (which could be grey, green or brown). I think some discussion of this and other mechanisms by which mental health benefits could be derived from natural spaces would be helpful here. Attention Restoration Theory is mentioned in the Discussion but could be introduced here. The nature connectedness literature could also be explored.
Thank you very much for this observation. Indeed by introducing more research associated with other mechanisms, particularly with restoration theory the manuscript improves its coherence. We have added new references explaining how nature is explained to be a potential positive exposure under restoration theory. Please see 67-70
- Line 59. “studying this association….” Please explain this point more clearly. Not sure what is meant by “lower boundaries”
We added a new sentence to explain what we mean by lower boundaries. Please see lines 61-62
- Line 100.When was the survey deployed? How does this match with when NDVI data were derived from? What are the implications of this?
We thank the reviewer for pointing out the need to clarify this issue as it is central to the interpretation of the results. The survey was deployed between May 2015 and December 2018. NDVI values were obtained for July 2016, which is mid-summer and expected to represent the greenest season in El Paso. The NDVI values were considered as representative of the long-term exposure to greenness by the participants. This was considered reasonable as the diversity of vegetation in the dessert region of El Paso is very small and hence the spatial variation of greenness as well. In other words, if a participant had a high NDVI value July 2016, that person is expected to be exposed to higher levels of greenness -at home- throughout the year, than someone that lived in a part of El Paso with lower NDVI values in July 2016. From this perspective, we were asking what is the association between long-term of exposure to residential greenness and depression severity among young local residents at a given common time in time (e.g., first semester of their nursing program). The assumptions have several limitations. For instance, there is definitely the possibility of that spatial variability of greenness might have seasonal changes, that might affect the interpersonal assessment of exposure to greenness. Similarly, exposure to greenness away from home would also affect the exposure assessment. Hence, our results should be considered as preliminary and more research needs to be conducted with more comprehensive longitudinal assessment of exposure to confirm our findings. We have added this discussion to the limitation section. Please see lines 383-397.
- Line 128. Some justification needed as to why this is an appropriate measure of “brownness”. What types of land use/cover does this include? You mention above “arid” natural spaces but could it also include some types of manmade spaces?
We have added a justification including in the appendix some samples of the images. Please refer to lines 136-146 in the manuscript and Appendix G in the supplementary material
- Line 192. Greenness missing an ‘s’
That was changed. Thanks.
- Line 194. Does this mean that non-residential areas within the buffers were excluded from the calculations?
Non-residential areas were not excluded from the calculations
- Line 195. Some guidance on how to interpret IRR values and what they represent would be useful.
We added a full explanation on how to interpret IRR values and what they represent. We appreciate this suggestion since before we had not mentioned in the methods section why we were using this measure of association. Please see lines 191-194.
- Figures 1-3. Some problems with the formatting of the figures – alignment issues and blank spaces appear over different areas of the figures obscuring the text. What is the x axis? IRR? Again, some guidance on what these values mean would be helpful. Denote as (a) continuous and (b) ordinal variables for clarity. I’m not sure what the diamond shapes represent. Figure 3 legend should say brownness not greyness.
Thanks we have reformatted figures. In the text we explain what the x axis represent IRR. We have denoted a) and b) for clarity. And changed to brownness Figure 3.
- Figures 4-6. Again some formatting problems as explanation of IRR would be helpful. Could the 3 separate figures be consolidated into one to avoid repetition, save space and allow for easier comparisons?
We have merged the 3 figures into one. We have now the categorical variable depression results in Appendix F. Results of this part are now much more succinct
- Line 336. Greyness not grayness (and in other places in the manuscript). As mentioned above, a description of the land cover types the brownness measure represents would be helpful. This would help link with the arguments related to spending time in natural environments below.
We fixed the greyness/grayness throughout the manuscript.
- Line 351. Or is preserving greenness less pressing if, as you have shown, brown spaces could provide similar benefits?
This is a tempting conclusion. However as we mention in reference to the low level of greenness in the second paragraph, we prefer to refrain of drawing more direct conclusions between greenness and depression.
- Line 365. How could the natural rock included in the greyness category affect this interpretation?
Please see our response in lines 343-346 in the Discussion section.
- Line 400. I would have liked to have seen a broader discussion of the many avenues this research could go in, specifically unpicking the mechanisms by which green and/or natural environments provide mental health benefits.
Thank you we have added some thoughts and reflections regarding these lines of research. Please see lines 403-411.
Round 2
Reviewer 1 Report
Thanks to the authors' effort, I believe this paper becomes stronger and more solid than before.
I agree with the publication of this paper.